# The Role of Oxidoreductase-Like Protein Olp1 in Sexual Reproduction and Virulence of *Cryptococcus neoformans*

**DOI:** 10.3390/microorganisms8111730

**Published:** 2020-11-04

**Authors:** Qi-Kun Yu, Lian-Tao Han, Yu-Juan Wu, Tong-Bao Liu

**Affiliations:** 1State Key Laboratory of Silkworm Genome Biology, Southwest University, Chongqing 400715, China; qikunyu@email.swu.edu.cn (Q.-K.Y.); hlt892996713@email.swu.edu.cn (L.-T.H.); wyj0213@email.swu.edu.cn (Y.-J.W.); 2Chongqing Key Laboratory of Microsporidia Infection and Control, Southwest University, Chongqing 400715, China

**Keywords:** *Cryptococcus neoformans*, oxidoreductases-like protein, Olp1, sexual reproduction, virulence

## Abstract

*Cryptococcus neoformans* is a basidiomycete human fungal pathogen causing lethal meningoencephalitis, mainly in immunocompromised patients. Oxidoreductases are a class of enzymes that catalyze redox, playing a crucial role in biochemical reactions. In this study, we identified one *Cryptococcus* oxidoreductase-like protein-encoding gene *OLP1* and investigated its role in the sexual reproduction and virulence of *C. neoformans*. Gene expression patterns analysis showed that the *OLP1* gene was expressed in each developmental stage of *Cryptococcus*, and the Olp1 protein was located in the cytoplasm of *Cryptococcus* cells. Although it produced normal major virulence factors such as melanin and capsule, the *olp1*Δ mutants showed growth defects on the yeast extract peptone dextrose (YPD) medium supplemented with lithium chloride (LiCl) and 5-fluorocytosine (5-FC). The fungal mating analysis showed that Olp1 is also essential for fungal sexual reproduction, as *olp1*Δ mutants show significant defects in hyphae growth and basidiospores production during bisexual reproduction. The fungal nuclei imaging showed that during the bilateral mating of *olp1*Δ mutants, the nuclei failed to undergo meiosis after fusion in the basidia, indicating that Olp1 is crucial for regulating meiosis during mating. Moreover, Olp1 was also found to be required for fungal virulence in *C. neoformans*, as the *olp1*Δ mutants showed significant virulence attenuation in a murine inhalation model. In conclusion, our results showed that the oxidoreductase-like protein Olp1 is required for both fungal sexual reproduction and virulence in *C. neoformans*.

## 1. Introduction

*Cryptococcus neoformans* is a globally distributed fungal pathogen causing life-threatening cryptococcal meningitis in both immunocompromised and immunocompetent patients [1]. An updated analysis recently showed that *C. neoformans* could cause nearly 300,000 infections and almost 200,000 deaths worldwide each year [2,3]. Initially, *C. neoformans* was proposed to contain three varieties, namely, *C. neoformans var. grubii*, *C. neoformans* var. *neoformans*, and *C. neoformans var. gattii* [4]. In 2002, the variety *C. neoformans var. gattii* was classified as an independent species *C. gattii* based on improved molecular methods [5]. Recently, some researchers proposed to recognize the two varieties of current *C. neoformans* (*C. neoformans* var. *grubii* and *C. neoformans* var. *neoformans*) as two separate species (*C. neoformans* and *C. deneoformans*) and split *C. gattii* into a total of five species (*C. gattii, C. bacillisporus, C. deuterogattii, C. tetragattii,* and *C. decagattii*) [6]. However, others recommend using the “*C. neoformans* species complex” and “*C. gattii* species complex” to recognize genetic diversity without creating the nomenclatural instability [7]. 

As a basidiomycetes yeast fungus, *C. neoformans* has two mating types, α and **a**, and it can undergo a dimorphic transition from yeast form to a filamentous growth form by mating and monokaryotic fruiting [8]. During mating in *C. neoformans*, after the fusion of the haploid yeast cells of the opposite mating types, dikaryotic filaments are formed, and a specialized sporulation structure called the basidium is eventually produced at the tip of the filaments. Next, four chains of basidiospores are produced on top of the basidium following the completion of meiosis inside the basidium [9]. Under certain conditions, the haploid cells of the same mating type of *C. neoformans*, e.g., α cells, can also fuse and undergo monokaryotic fruiting to generate filaments and basidiospores [10]. The major difference between mating and fruiting is that the hyphal cells produced during mating contain two nuclei and are linked by fused clamp connections, whereas those produced during fruiting are mononucleate with unfused clamp connections [10]. 

As an important human fungal pathogen, *C. neoformans* has three classical virulence factors: capsule formation, melanin production, and growth at 37 °C [11,12]. Other virulence factors such as urease [13], extracellular phospholipase activity [14], mannitol production [15,16], laccase [17], hyaluronic acid [18], calcineurin [19], and metalloprotease [20,21] also contribute to the infection and the pathogenesis of *C. neoformans*. However, these studies indicate that fungal virulence is a complex trait, and the determining mechanism of virulence still needs to be further explored.

Oxidoreductases are a class of enzymes known as dehydrogenases or oxidases that catalyze redox reactions, including dehydrogenases, oxygenases, oxidases, and reductases [22]. Redox reactions involve many basic metabolism processes such as glycolysis, the tricarboxylic acid (TCA) cycle, oxidative phosphorylation, and amino acid metabolism, which are essential for life. The dysfunction of oxidoreductases can cause many diseases in humans, such as cancer [23,24,25,26], neurodegenerative disorders [27,28], and cardiovascular disease [29,30]. Oxidoreductases have also been reported to be extremely critical to fungi. In the plant pathogenic fungus *Magnaporthe oryzae*, dehydrogenase *MoSFA1* is required for conidiation and contributes to virulence in the penetration and biotrophic phases [31]. In the entomogenous fungus *Beauveria bassiana*, fungal benzoquinone oxidoreductase is a host-specific virulence factor and is beneficial for the fungi to resist the benzoquinone exposure [32]. In *Candida albicans*, oxidoreductase FLPs reduce ubiquinone (coenzyme Q), which can serve as an antioxidant in the membrane to protect *Candida albicans* from oxidative stress and promote the fungal virulence [33]. In *Aspergillus fumigatus*, a fungal alcohol dehydrogenase required for the last step of ethanol fermentation in response to hypoxia is critical to pulmonary invasive fungal infection [34]. Moreover, oxidoreductases including alternative oxidase [35], cytochrome c peroxidase [36], mannitol-1-phosphate-5-dehydrogenase [37], thiol peroxidase [38], and thioredoxin reductase [39] already have been confirmed to be extremely significant for the virulence of *C. neoformans* [40]. 

In the present study, we have identified and functionally analyzed one oxidoreductase-like protein Olp1 in *C. neoformans*. Sequence analysis revealed that Olp1 does not contain any currently known domain and had low homology with other non-*Cryptococcus* proteins. Expression analysis showed that the *OLP1* gene was down-regulated during mating and the Olp1 protein was localized in the cytoplasm of cryptococcal cells. Growth under stress conditions showed that the *olp1*Δ mutants were hypersensitive to LiCl and 5-FC, although the mutants produced normal classical virulence factors. We also found that Olp1 was crucial for sexual reproduction, as the *olp1*Δ mutants produced short and sparse dikaryotic hyphae and no basidiospores in bilateral mating. Moreover, Olp1 is also required for fungal virulence, as the *olp1*Δ mutants showed a significant virulence attenuation in the mouse inhalation model. Overall, our results suggest that Olp1 regulates sexual reproduction and fungal virulence in *C. neoformans*.

## 2. Materials and Methods

### 2.1. Ethics Statement

The animal studies conducted at Southwest University were in full compliance with the “Guidelines on Ethical Treatment of Experimental Animals (2006, No. 398)” issued by the Ministry of Science and Technology of China and the “Regulation on the Management of Experimental Animals (2006, No. 195)” issued by Chongqing Municipal People’s Government. The Animal Ethics Committee of Southwest University approved all of the vertebrate studies on 6 March 2019 with a project identification code IACUC-20190306-07. 

### 2.2. Strains, Media, and Growth Conditions

The wild-type *Cryptococcus* strains and their derivatives used in the present study are shown in Table 1. Yeast extract peptone dextrose (YPD) agar medium was used for the routine cultivation of *Cryptococcus* strains at 30 °C. Murashige and Skoog medium (MS) and V8 medium used for *Cryptococcus* mating induction and sporulation assays were prepared as described previously [41]. All other media were prepared as described previously [42].

### 2.3. OLP1 Gene Expression Pattern Analysis

To understand the temporal expression pattern of the *OLP1* gene and subcellular localization of the Olp1 protein, a *Hind*III/*BamH*I genomic DNA fragment containing a 1.5-Kb upstream promoter region, the open reading frame (ORF) of the *OLP1* gene without the stop codon, was amplified using primers TL727/TL729 (see Table 2 for primer sequences) and cloned into pTBL3 [45] to generate the *P_OLP1_-OLP1-mCherry* fusion plasmid pTBL118 (see Table 1 for plasmid information) for Olp1-mCherry fusion proteins. Then, the *Sca*I linearized pTBL118 was concentrated and biolistically transformed into both α and **a** mating-type *olp1*Δ mutant strains (TBL337 and TBL348), respectively. Stable transformants were selected on YPD medium containing nourseothricin sulfate (100 mg/liter). The fluorescence of the transformants was examined using an Olympus inverted confocal laser scanning microscope (Olympus, FV1200).

### 2.4. Detection of OLP1 Gene Expression Using qRT-PCR

The expression of *OLP1* during mating was also tested by qRT-PCR. Cell cultures preparation, mating mixture collection, RNAs extraction, cDNAs synthesis, and specific methods of Realtime PCRs are the same as previously described [42,45]. The expression levels of the *GAPDH* gene were used as an internal control. Primers TL217/TL218 and TL539/TL540 were used to amplify the *GAPDH* gene and *OLP1* gene, respectively. 

### 2.5. Generation of OLP1 Gene Knockout, Complementation, and Overexpression Strains

The *olp1*Δ mutant strains were generated in both H99 and KN99a strain backgrounds using the split marker strategy, as described previously [45,47]. Primers used to amplify the overlapping PCR fragments for *OLP1* gene knockout are listed in Table 2. Stable transformants were further confirmed on YPD medium containing antibiotic G418 (200 mg/liter). To screen the *olp1*Δ mutants, diagnostic PCR was used by analyzing the 5′ junction of the disrupted mutant alleles with positive primers F4/R4 (TL403 and TL59) and negative primers F3/R3 (TL1255 and TL1256). Positive transformants identified by PCR screening were further confirmed by Southern blot analysis.

To generate the *olp1*Δ mutant complementation strains, a 3.9 Kb genomic DNA fragment that contains the upstream promoter region, the ORF of the *OLP1* gene, and its terminator region was amplified with primers TL556/TL1257 and cloned into the pTBL1 [45] vector to generate an *OLP1* gene complementation vector pTBL93. The pTBL93 plasmid was linearized by *Sca*I and biolistically transformed in both α and a mating-type *olp1*Δ mutants strains. Lithium chloride (LiCl) sensitivity assay was performed to identify transformants that complemented the *olp1*Δ phenotype.

To construct the *OLP1* gene overexpression *OLP1*^OE^ strains, we amplified a DNA fragment (1.4 Kb) containing the coding region of the *OLP1* gene with primers TL515/TL516 and cloned it into an expression vector pTBL5 [45] controlled by the *ACTIN* promoter to generate an *OLP1* gene overexpression vector pTBL85. Then, the resulting vector, pTBL85, was linearized by *Sca*I and biolistically transformed in both α and **a** mating-type *olp1*Δ mutants strains. Stable transformants were selected on YPD medium containing Nourseothricin Sulfate (100 mg/liter). The *OLP1*^OE^ strains were further confirmed by fluorescence observation and quantitative real-time PCR.

### 2.6. Generation of Nop1-mCherry Strains

To monitor the nuclear positioning in *Cryptococcus*, we digested a previously constructed plasmid pTBL68 [45] with *Sac*II and *Nde*I to generate the *NOP1-mCherry-NAT* fragments and biolistically transformed the fragments into α and **a** mating-type strains of the *olp1*Δ mutants. The native *NOP1* gene was replaced with the *NOP1-mCherry-NAT* cassette by homologous recombination. Positive transformants were further confirmed by PCR to confirm the homologous recombination, and the nuclear signal of the fluorescent mCherry was screened by Olympus inverted confocal laser scanning microscopy (Olympus, FV1200).

### 2.7. Assays for Melanin, Capsule Production, and Mating

To test the role of Olp1 protein in melanin and capsule production in *Cryptococcus*, yeast cells of H99, *olp1*Δ mutants, *olp1*Δ::*OLP1* complementation strains, and *OLP1*^OE^ overexpression strains were induced on Niger seed agar medium and Dulbecco’s modified Eagle medium (DMEM), respectively. In the mating assay, cell suspensions of opposite mating types (α or a) of each of the above strains were mixed and cocultured on MS or V8 agar medium at 25 °C in the dark. The matings between the α and **a** Nop1-mCherry strains of wild-type strains (TBL101 × TBL102) or *olp1*Δ mutants (TBL371 × TBL372) were also induced on MS media, respectively, and their fungal nuclei development were monitored during the whole mating process. The detailed methods were described previously [42,45].

### 2.8. Virulence Studies

To examine the role of the Olp1 protein in fungal virulence, we washed the overnight cultures of each yeast strain with PBS buffer and resuspended the cultures to a final concentration of 2 × 10^6^ cells/mL. Female C57 BL/6 mice (10 mice of each group) were intranasally infected with 10^5^ cells of each yeast strain, as described previously [13]. Animals that appeared moribund or in pain were sacrificed by CO_2_ inhalation. Survival data and fungal burden between paired groups were statistically analyzed with PRISM version 8.0 (GraphPad Software, San Diego, CA, USA) (*p* values of <0.05 were considered significant) as previously described [45].

### 2.9. Histopathology and Fungal Burdens in Infected Organs

Infected mice were sacrificed at the endpoint of the experiment according to Southwest University-approved animal protocol. The isolation of the infected brains, lungs, and spleens and the preparation of the tissue slides were the same as previously described [45]. Tissue slides were stained with hematoxylin and eosin (H&E) and examined by light microscopy (Olympus, BX53, Tokyo, Japan). The brains, lungs, and spleens of infected mice were dissected and homogenized in PBS buffer. Then, 100 µl of each diluted resuspension was spread on YPD plates containing ampicillin and chloramphenicol and incubated at 30 °C for three days to determine the colonies [45].

## 3. Results

### 3.1. Identification of the Oxidoreductases-Like Protein Olp1 in C. neoformans

In our previous study on the function of autophagy-related proteins (Atgs), immunoprecipitation (IP) pulldown and liquid chromatography with tandem mass spectrometry (LC-MS/MS) analysis were used to identify the interacting proteins of Atgs. From LC-MS/MS data, one protein, CNAG_00472, was found to have potential interactions with Atg5, Atg8, and Atg12, respectively. In this article, we further studied the function of CNAG_00472 in *C. neoformans*. We first searched the CNAG_00472 gene in the FungiDB database (http://fungiDB.org) [48] and found that the CNAG_00472 gene encodes a protein of 412 amino acids with a predicted molecular mass of 72 KDa. Protein conservative domain analysis showed that this protein did not contain any currently known domain except for a glycosylphosphatidylinositol (GPI) anchor attach site (Figure 1A). Smart blast and phylogenetic analysis revealed that the predicted protein shows high sequence identities (94%) with the species inside the *Cryptococcus* genus but low sequence identities (22–27%) to several proteins, including putative oxidoreductases in *Saccharomyces cerevisiae*, *Neurospora crassa*, and *Candida glabrata*, dehydrogenases in *Schizosaccharomyces pombe* and *Magnaporthe oryzae* (Figure 1B). Therefore, we named this novel protein (CNAG_00472) Olp1 (oxidoreductase-like protein 1). Meanwhile, reciprocal blast analysis using yeast oxidoreductase (YNL181W) as a query was performed against the *C. neoformans* genome to ensure that the most similar sequence of cryptococcal Olp1 was the same as that of the *C. neoformans* inquiry gene (Figure 1C). These results suggested that the Olp1 protein might be an oxidoreductases-like protein. 

### 3.2. Expression of the OLP1 Gene in C. neoformans

To study the expression pattern of the *OLP1* gene in *C. neoformans*, we first detected the expression levels of the *OLP1* gene in different developmental stages using qRT-PCR. Mating mixtures of the wild-type strains (H99 × KN99**a**) were harvested from V8 plates after incubation for 0, 12, 24, 48, 72 h, and 7 d. RNAs were purified, cDNA was synthesized, and qRT-PCR was performed. The results of qRT-PCR showed that compared to the 0 h time point, the transcription level of the *OLP1* gene was down-regulated during mating, especially after 48 h, indicating that Olp1 may play a role in later stages of mating after cell fusion (Figure 2A).

To observe the expression of the *OLP1* gene more intuitively in different developmental stages in *C. neoformans*, we constructed the *Cryptococcus* strains expressing the Olp1-mCherry fusion proteins (TBL373 and TBL374, see Table 2) under the control of the *OLP1* gene native promoter. The strains (TBL373 and TBL374) expressing Olp1-mCherry showed no apparent difference in LiCl resistance than the wild-type strains, which indicated that Olp1-mCherry was fully functional (Appendix A). The integrity of the Olp1-mCherry fusion proteins was also confirmed by Western blotting (Appendix A). Mating of the Olp1-mCherry strains (TBL373 and TBL374) was induced on MS medium, and the fluorescence of Olp1-mCherry fusion protein in different developmental stages was observed by confocal microscopy. Fluorescence microscopy showed that the Olp1-mCherry fusion protein was expressed in all different developmental stages of *C. neoformans*, including yeast cells, dikaryotic hyphae, basidia, and basidiospores, and it was located in the cytoplasm of *Cryptococcus* cells, indicating that the Olp1 protein may play a role in the sexual reproduction of *C. neoformans* (Figure 2B).

We also tested the localization of Olp1-mCherry inside the yeast cell under different stress conditions such as high-temperature stress (37 °C), osmotic stress (1.5 M Sorbitol, 1.5 KCl, and 1.5 M NaCl), cell wall stress (0.025% SDS), oxidative stress (2.5 mM H2O2), nitrosative stress (1 mM NaNO_2_, pH = 4.0), and metal ions (LiCl). As a result, we found no difference between the localizations of Olp1-mCherry under the above stress conditions (Figure 2C).

### 3.3. Olp1 Plays Roles in Stress Responses

To evaluate the function of the Olp1 protein in *C. neoformans*, we constructed *olp1*Δ mutants (TBL337 and BL348) in both mating types of *C. neoformans* H99 strain backgrounds (Figure 3). The *olp1*Δ::*OLP1* complementation strains (TBL349 and TBL352) and *OLP1*^OE^ overexpressed strains (TBL353 and TBL354) were also obtained. The overexpression of *OLP1* was confirmed by qRT-PCR (Figure 4C). 

*C. neoformans* has three classical virulence factors: capsule formation, melanin production, and the ability to grow at mammalian body temperature. To investigate the role of Olp1 in fungal virulence in *C. neoformans*, we first examined the development of these virulence factors in the above strains we generated. However, there was no significant difference in the development of the capsule and melanin between wild-type, *olp1*Δ mutants, *OLP1* complemented, or overexpressed strains (Figure 4A,B) except that the *olp1*Δ mutants showed minor growth defects on YPD pates at 30 or 37 °C (Figure 4D). Notably, the growth of the above strains under stress conditions showed that the *olp1*Δ mutants were sensitive to 5-FC and LiCl but not sensitive to other tested stress conditions such as hydroxyurea (HU) and methyl methanesulfonate (MMS) (Figure 4D), indicating that the Olp1 protein may play an important role in RNA processing and synthesis or maintaining lithium-ion homeostasis and protecting fungi from lithium-ion toxicity.

### 3.4. Olp1 is Crucial for Sexual Sporulation

As a basidiomycetes fungus, *C. neoformans* has two mating types and can undergo sexual reproduction, which involves the fusion of haploid cells of the opposite mating type, α and **a**, to produce dikaryotic filaments and basidiospores. To investigate whether Olp1 plays a role in mating, we examined the matings of wild-type (H99 × KN99**a**), *olp1*Δ mutants (TBL337 × TBL348), *olp1*Δ::*OLP1* complementation strains (TBL349 × TBL352), and *OLP1*^OE^ overexpression strains (TBL353 × TBL354) on MS media. Interestingly, the bilateral mating between the *olp1*Δ mutants produced shorter and sparser dikaryotic hyphae and failed to generate basidiospores when compared with wild-type strains, while the *OLP1* complemented or overexpressed strains generated normal dikaryotic hyphae and basidiospores (Figure 5A). Therefore, our results demonstrate that the Olp1 protein is crucial for sexual reproduction in *C. neoformans*.

### 3.5. Olp1 Is Involved in Meiosis and Nuclear Division

To further explore why the *olp1*Δ mutants failed to produce basidiospores during mating, we fused the Nop1 nucleolar protein [49] with the fluorescent mCherry (C-terminus tagged) to monitor the fungal nuclei positioning at different stages of sexual reproduction in *C. neoformans*. A single nucleus can be observed in each yeast cell of both the wild-type (TBL101 or TBL102) and the *olp1*Δ mutants (TBL371 × TBL372) cultures (Figure 5B, first panel), and two separated nuclei were observed in each dikaryotic hypha and the young basidia produced from bilateral-mating mixtures of both the wild-type and *olp1*Δ mutants (Figure 5B, second and third panels). A single fused nucleus could be observed in the young basidium of both the wild-type and *olp1*Δ mutants, indicating that both strains undergo normal nuclear fusion to produce basidia during mating (Figure 5B, fourth panel). However, the nuclei of the *olp1*Δ mutants can not undergo meiosis after fusion in the bilateral mating, and only one nucleus could be visualized in each mature basidium after 14 days of incubation, while all basidia from wild-type mating produced four nuclei (Figure 5B, fifth panel). These results indicate that Olp1 plays a crucial role in the meiosis during the sporulation development stage in *C. neoformans*.

### 3.6. Olp1 Is Required for Fungal Virulence

To determine the role of the Olp1 protein in fungal virulence in *C. neoformans*, we tested the virulence of *olp1*Δ mutants in a murine inhalation model of cryptococcosis. Female C57BL/6 mice were infected intranasally with 10^5^ yeast cells of the wild-type, *olp1*Δ mutant, or complemented *olp1*Δ::*OLP1* strain and monitored for signs of morbidity. All mice infected by the wild-type and complemented strains were terminated between 22 and 27 days postinfection (DPI) (*p* > 0.999), while most of the mice infected by the *olp1*Δ mutants could survive between 58 and 75 DPI (*p* < 0.0001). There was still one mouse alive when we terminated the experiment after 80 days postinfection (Figure 6A). Hence, the virulence attenuation observed in the *olp1*Δ mutant was due to the deletion of the *OLP1* gene. 

To further explore why the *olp1*Δ mutants have virulence defects, we examined the fungal burdens in the organs of the infected mice at the endpoint of the infection experiments. Brains, lungs, and spleens from five mice infected by each strain were isolated, and the fungal burden of these organs was evaluated as yeast colony-forming units (CFUs) per gram of fresh organ. Mice that were infected by *olp1*Δ mutants exhibited lower fungal burdens in the lungs and spleens but higher fungal burdens in the brains than those of mice infected with the wild-type strain (Figure 6B) (*p* < 0.0001, *p* < 0.01, and *p* < 0.01, respectively). These data showed that the virulence attenuation in *olp1*Δ mutants was due to the disruption of the *OLP1* gene. 

Fungal lesion development in brains, lungs, and spleens was also visualized in H&E-stained slides. Both the wild-type and the complemented *olp1*Δ::*OLP1* strains caused severe lesions in infected brains, lungs, and spleens at 24 DPI (Figure 6C, top, and bottom panels). Meanwhile, a large number of yeast cells with thickened capsules can be easily observed in all of these three organs (Figure 6C, top, and bottom panels). In contrast, the *olp1*Δ mutants strain only can cause very limited lesions in the infected lungs and spleens, with very few yeast cells observed at the end time point (Figure 6C, middle panel). However, the *olp1*Δ mutants strain can cause severe lesions in the brain with numerous yeast cells observed at the end time point (Figure 6C), suggesting that the mortality observed in mice infected with the *olp1*Δ mutants was due to central nervous system (CNS) disease. Taken together, we showed that Olp1 is required for the fungal virulence of *C. neoformans*. 

### 3.7. Olp1 Is Vital for Fungal Infection Progression

To better understand the *olp1*Δ mutant–host interaction dynamics during the infection progression, fungal burdens in organs were examined, and fungal lesions development was also visualized in H&E-stained slides at 7, 14, and 21 DPI. 

At 7 DPI, there were no cryptococcal yeast cells recovered from the brains of the mice infected with the wild-type or *olp1*Δ mutants, and the brains remain intact (Figure 7A,D). At 14 DPI, yeast cells can be recovered from the brains of the mice infected with both the wild-type and *olp1*Δ mutant strains. However, the CFU number recovered from the brains infected by *olp1*Δ mutants is much less than that recovered from the wild type (Figure 7A). Notably, at 21 DPI, the CFU number recovered from the brains infected by the wild type was significantly higher than that of the *olp1*Δ mutants (Figure 7A) (*p* = 0.004). Meanwhile, the mice infected with the wild type had suffered cryptococcosis, and their brains had been severely damaged, while the *olp1*Δ mutant-infected mice were healthy, and no lesions were found in their brains (Figure 7D). The fungal burdens in the lungs of the mice infected by the wild-type strain are significantly higher in comparison with that of the mice infected by the *olp1*Δ mutants at 7, 14, and 21 DP1 (Figure 7B) (*p* = 0.01, *p* = 0.003, and *p* = 0.0008, respectively). Interestingly, the fungal burdens (≈10^3^) in the lungs of the mice infected by *olp1*Δ mutants decreased gradually over time after infection (Figure 7B). The lung damage caused by wild-type infection worsened with the constant proliferation of cryptococcal cells, while the lungs infected by *olp1*Δ mutant remained relatively intact (Figure 7E). The fungal burdens in the spleens of wild-type infected mice were also higher than that of the *olp1*Δ mutant-infected mice (Figure 7C) (*p* = 0.03). Observation of H&E-stained slides also showed the accumulation of cryptococcal cells intensively in the spleen of wild-type infected mice at 21 DPI, while no yeast cell could be observed from the slides of the *olp1*Δ mutants (Figure 7F).

Taken together, our results indicate that the Olp1 protein plays a significant role in the development of meningitis in a murine model.

## 4. Discussion

Oxidoreductases are a large group of enzymes that play an indispensable role in the primary metabolism of life. In this paper, we identified an oxidoreductase-like protein named Olp1 and proved that it is required for sexual reproduction and fungal virulence in *C. neoformans*. The *olp1*Δ mutants produced short and sparse dikaryotic hyphae and failed to produce infectious basidiospores in a bilateral mating assay. Fungi nuclei development assay showed that two nuclei in the basidia of *olp1*Δ mutants could fuse but were unable to undergo meiosis, which suggested that the Olp1 protein was crucial for regulating meiosis during the sexual reproduction. Notably, the *olp1*Δ mutants showed significant virulence attenuation in a murine model, implying that the Olp1 protein may play an important role in fungal virulence in *C. neoformans*.

Virulence factor assay in this study showed that Olp1 was not involved in canonical virulence factors production such as melanin production, capsule formation, and growth at mammalian body temperature. However, the *olp1*Δ mutants exhibited growth defects on the YPD medium supplemented with LiCl, suggesting that Olp1 is vital to maintain the lithium-ion homeostasis and to protect the cell from the toxicity of lithium-ion. Lithium salts have been used as a mood-stabilizing agent for the treatment of bipolar disorder for a long time, and lithium-ion can have a profound effect on both human behavior and early embryonic development. The mechanism of action of lithium was extraordinarily complex, and some research showed that LiCl had numerous targets such as inositol monophosphatases [50], bisphosphate 3-nucleotidase [51], and glycogen synthase kinase-3 (GSK-3) [52]. There are two identified targets of LiCl in *S. cerevisiae*: RNA processing enzymes [53] and HAL2 Nucleotidase [54]. In *S. cerevisiae*, it was reported that the halotolerant protein kinase Hal5p was a high-copy suppressor, which can inhibit nearly one-third of the identified sporulation and meiosis-related lithium-sensitive mutant genes [55]. Our study showed that Olp1 was required for cell growth on the YPD medium supplemented with LiCl, meiosis and fungal sporulation in *C. neoformans*. Whether Olp1 is under the regulation of Hal5p is an exciting subject that needs to be further investigated.

Our study also showed that the *olp1*Δ mutants are sensitive to the antifungal drug 5-fluorocytosine (5-FC) (Figure 4C). 5-FC, one of the oldest antifungal agents, can be transported and converted into toxic 5-fluorouracil (5-FU) by cytosine deaminase within the susceptible fungal cell. The metabolites of 5-FU can both inhibit fungal RNA processing and synthesis as well as DNA synthesis by incorporating into them [56]. 5-FC not only inhibits the processing of pre-rRNA into mature rRNA [57] but also disrupts the post-transcriptional modification of tRNAs [58] and the assembly and activity of snRNA/protein complexes, thereby inhibiting the splicing of pre-mRNA [59]. Our result showed that the *olp1*Δ mutant exhibited susceptibility to 5-FC but not to the DNA-damaging agents such as hydroxyurea (HU) [60] and methyl methanesulfonate (MMS) (Figure 4D) [61]. Thus, we speculated that the Olp1 protein plays an essential role in RNA processing and synthesis. However, further investigation is still needed to determine the exact molecular mechanism of this phenotype. 

The putative oxidoreductase encoding gene *PBR1* is required for cell viability as the *PBR1* deletion strain is inviable in *S. cerevisiae*, indicating that the oxidoreductase is essential for *S. cerevisiae* survival [62,63,64]. However, although the *olp1*Δ mutants showed sexual reproduction defects and significant attenuation in fungal virulence, the *olp1*Δ mutant cells survived and even appeared healthy under most of the tested growth conditions. Are there any other putative oxidoreductase-like proteins in the *C. neoformans* genome? To answer this question, we performed a reciprocal blast analysis using yeast oxidoreductase (YNL181W) as a query against the *C. neoformans* genome, and the reciprocal blast results showed that there are at least nine more oxidoreductase-like proteins except for the Olp1 (Figure 1C). Perhaps due to the functional redundancy caused by the presence of multiple oxidoreductase-like genes in *C. neoformans* genome, the *olp1*Δ mutant could still survive and even appear to be healthy in most of the tested growth conditions.

Fungal sporulation is an extremely delicate strategy utilized by fungi to reproduce, disseminate, and survive [65,66]. Surprisingly, our results suggested that the Olp1 protein played an essential role in fungal sporulation, since the mating of the *olp1*Δ mutants produced shorter and sparser hyphae and failed to produce basidiospores. The fungal nuclei development assay in our study showed that the *olp1*Δ mutants failed to undergo meiosis after nuclei fusion during mating, which demonstrated that Olp1-mediated homeostasis played an essential role in the regulation of the sporulation and meiosis process. However, why the oxidoreductase-like protein Olp1 is required for the sexual reproduction of *C. neoformans*, especially the meiotic process of sexual reproduction, is challenging to answer for the time being. So far, there have been reports about oxidase affecting fungal sexual reproduction. In *Aspergillus nidulans*, the nicotinamide adenine dinucleotide phosphate (NADPH) oxidase noxA proved to play an essential role in the sexual fruit body differentiation by affecting the generation of the reactive oxygen species (ROS) [67]. In *Neurospora crassa*, the NADPH oxidase Nox1 requires the regulatory subunit NOR-1 to control the sexual cell differentiation [68]. The NADPH oxidases regulatory subunit FgNoxR is essential for sexual development in F*usarium graminearum* [69]. The above three examples all illustrated the role of reactive oxygen species produced by oxidase in fungal sexual reproduction regulation. As an oxidoreductase-like protein, the Olp1 may also play an essential role in regulating of ROS production and then affect the sexual reproduction process in *C. neoformans* through disruption of one of the most critical meiosis-specific proteins during the sporulation. However, whether the absence of the Olp1 protein affects the ROS production or meiosis-related proteins involved in sporulation remains unknown and needs to be further studied.

The deletion of the *OLP1* gene resulted in significant fungal virulence attenuation in *C. neoformans*. Fungal burdens at the endpoint showed that the *olp1*Δ mutants accumulated significantly inside the brains but not in the lungs and spleens. Meanwhile, the *olp1*Δ mutants need more time (>60 days) than the wild-type strain to reach a similar fungal burden inside the brains at the end of the infection, which might be caused by the different growth rates of the *olp1*∆ mutants and wild-type strains in the brains. The mammalian brains contain more abundant inositol than that in the plasma [70], which helps *Cryptococcus* grow inside the brains, since *Cryptococcus* is the only fungus that can use inositol as a sole carbon source [71,72]. Our previous study showed that brain inositol is a novel stimulator for promoting *Cryptococcus* penetration of the blood–brain barrier [73]. Once in the brain, *Cryptococcus* can use inositol in the brain for rapid growth. However, in this study, the *olp1*∆ mutants need more time than the wild-type strain to reach a similar fungal burden at the end of the infection, suggesting that the *olp1*∆ mutants may grow slower than the wild-type strain inside the brain, which is possibly due to a defect in inositol utilization. The metabolism of inositol involves a variety of oxidoreductase, and Olp1, as an oxidoreductase, might be involved in the metabolism of inositol. Therefore, deletion of the *OLP1* gene might affect the metabolism of inositol in *Cryptococcus* and lead to the slow growth of the *olp1*∆ mutants in the mouse brain, which results in a longer time for the *olp1*∆ mutants to reach a similar fungal burden at the end of the infection than the wild-type strain. However, whether the *olp1*∆ mutants have defects in the utilization of inositol needs to be further investigated.

The growth rate of *olp1*Δ mutants in mouse lung and spleen tissues was significantly lower than that of wild-type strains. The fungal burdens in the lungs of the *olp1*Δ mutants-infected mice were at a low level at each time point (≈10^3^). One possible explanation is that the microenvironment of the lung or spleens is more hostile than that of the brain and is not suitable for *olp1*Δ mutants survival, which also explains that the Olp1 protein is vital for fungal virulence in *C. neoformans*. Meanwhile, our results also suggested that the disturbance of ion homeostasis and abnormal RNA processing and synthesis may be possible reasons for the virulence defect of *olp1*Δ mutants in the murine inhalation model of infection. However, further study is still needed to investigate the specific mechanism of fungal virulence attenuation in *olp1*Δ mutants.

In conclusion, this report describes the identification and characterization of an oxidoreductase-like protein Olp1 in *C. neoformans*. The functional study of this protein reveals that the Olp1 protein is involved in the regulation of sexual reproduction and virulence in *C. neoformans*. This protein may be involved in RNA processing and synthesis; it may also play a key role in maintaining lithium-ion homeostasis and protecting fungi from lithium-ion toxicity.

## Figures and Tables

**Figure 1 microorganisms-08-01730-f001:**
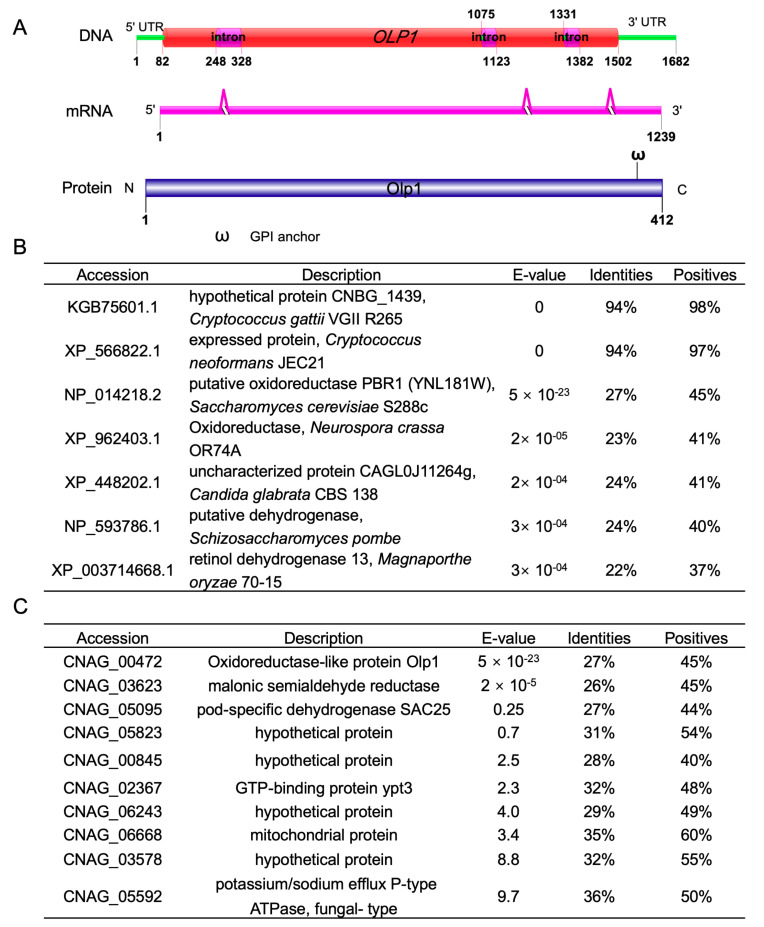
Sequence analysis of the oxidoreductase-like protein Olp1. (**A**) Schematic diagram of the genomic DNA, mRNA, and protein of Olp1. (**B**) The sequences alignment result of Olp1 and its homologs. (**C**) Reciprocal blast analysis of yeast oxidoreductase (YNL181W) against the *C. neoformans* genome.

**Figure 2 microorganisms-08-01730-f002:**
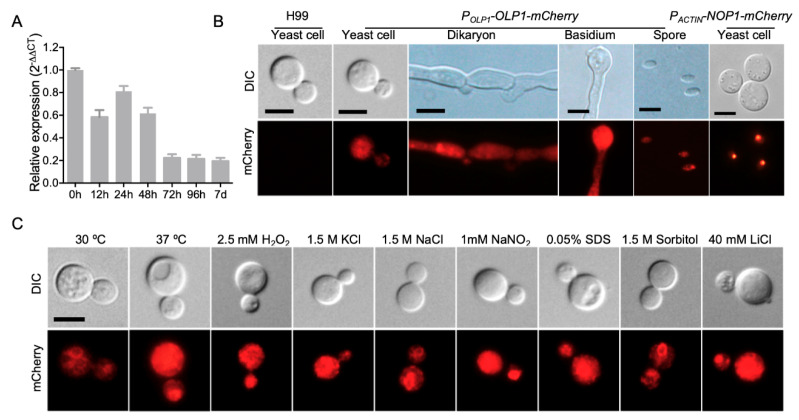
Expression pattern of the *OLP1* gene in *C. neoformans*. (**A**) The expression of *OLP1* during mating on V8 medium was measured by qRT-PCR. Mating mixtures between H99 and KN99**a** were harvested from V8 plates after incubation for 0, 12, 24, 48, 72 h, and 7 d. RNAs were purified, cDNA was synthesized, and qRT-PCR was performed. The comparative C_T_ method was used for the relative quantification, and the *GAPDH* gene was used as an endogenous reference. The error bars show standard deviations of three repeats. (**B**) Expression and localization of the Olp1-mCherry fusion protein in various development stages of *C. neoformans*. Representative bright-field and fluorescence images of the yeast cell, dikaryon, basidium, and spores are shown. (**C**) Subcellular localization of Olp1-mCherry (under control of *OLP1* native promoter) in yeast cells under different stress conditions. DIC: differential interference contrast; Bar = 5 μm.

**Figure 3 microorganisms-08-01730-f003:**
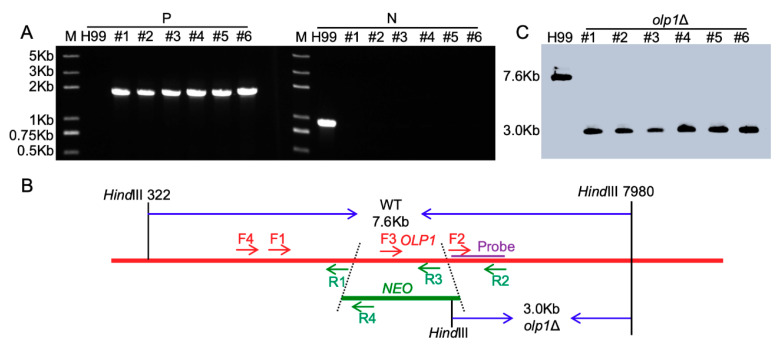
Generation of the *olp1*Δ mutants. (**A**) PCR verification of *OLP1*-disrupted transformants; H99: positive control using the wild-type genomic DNA; #1–6: Six G418 resistant transformants used for PCR screening; P: Positive primers (TL410/TL59), F4/R4 in 3B; N: Negative primers (TL1255/TL1256), F3/R3 in 3B; M: 2K Plus DNA Ladder. (**B**) Restriction enzymes used for digestion of the genomic DNAs for Southern blot. WT: wild-type strain genomic DNA; *olp1*Δ: *olp1*Δ mutant genomic DNA; *NEO*: the G418 resistant marker; The TL1253/TL1254 (F2/R2) PCR products were used as templates to synthesize the probe. The wild-type strain H99 will generate a 7.6-Kb band, while the *olp1*Δ mutants generate a 3.0-Kb band. (**C**) Southern blot analysis of the *OLP1* disrupted transformants. All genomic DNAs of the six G418 resistant transformants were digested with *Hind*III, fractionated, and hybridized with a probe located in the downstream flanking sequence of *OLP1* shown in Figure 3B. As expected, a 3.0-Kb band was detected in *olp1*Δ mutants in contrast with a 7.6-Kb band in the wild-type strain H99.

**Figure 4 microorganisms-08-01730-f004:**
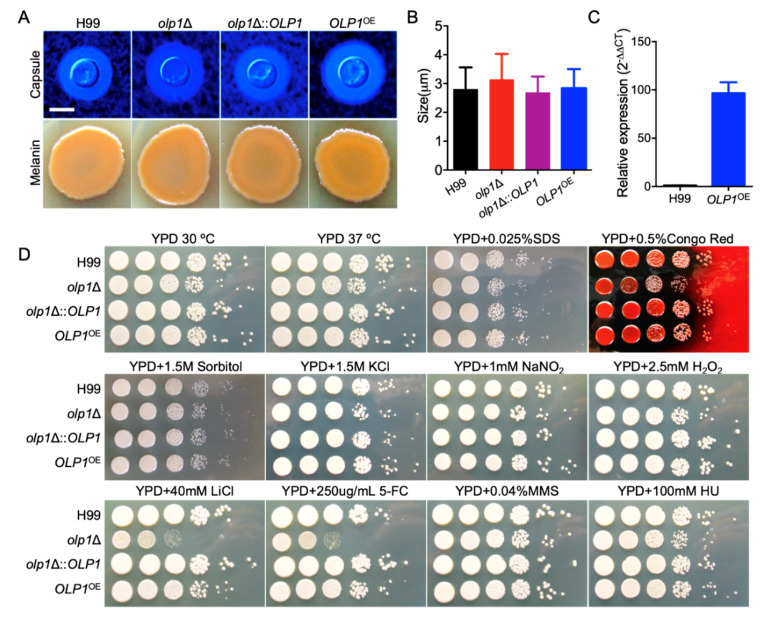
The *olp1*Δ mutants are sensitive to 5-fluorocytosine (5-FC) and lithium chloride (LiCl). (**A**) Capsule formation (top) and melanin production (bottom) of each cryptococcal strain. (**B**) Statistical analysis of the capsule formation in each *Cryptococcus* strain. (**C**) The overexpression of the *OLP1* gene was measured by relative qRT-PCR analysis. (**D**) Growth of each cryptococcal strains on yeast extract peptone dextrose (YPD) or YPD supplemented with various stress reagents. The plates were grown for two days at 30 °C. The name of the strains is indicated on the left and the conditions are indicated at the top.

**Figure 5 microorganisms-08-01730-f005:**
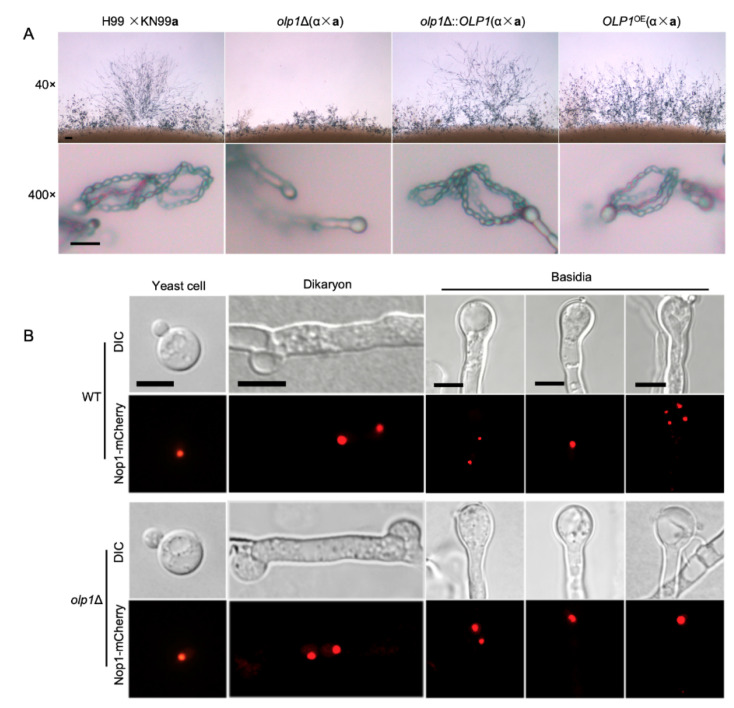
Mating hyphae production and sporulation on the wild-type and *olp1*Δ mutants. (**A**) Induction of the mating of wild-type (H99 × KN99**a**), *olp1*Δ mutants (TBL337 × TBL348), *olp1*Δ::*OLP1* complementation strains (TBL349 × TBL352), and *OLP1^OE^* overexpression strains (TBL353 × TBL354) on MS media. Mating structures at 40× magnification (top) and 400× magnification (bottom) were incubated at 25 °C in the dark before being photographed. Bar = 10 μm. (**B**) The development of fungal nuclei in yeast cells, mating hyphae, and basidia of the wild-type (TBL101 × TBL102, top) and *olp1*Δ mutants (TBL371 × TBL372, bottom). After incubating on MS medium in the dark for 7 or 14 days, the mating cultures were isolated and visualized by confocal microscopy (Olympus, FV1200). DIC: differential interference contrast; Bars, 5 µm.

**Figure 6 microorganisms-08-01730-f006:**
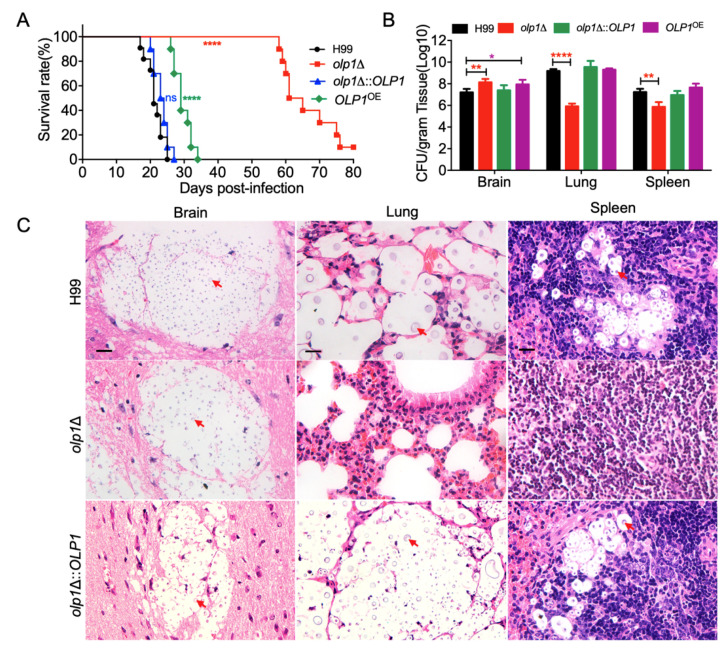
Olp1 regulates fungal virulence in a mouse systematic infection model. (**A**) Female C57 BL/6 mice were infected intranasally with 10^5^ cells of the wild-type H99, the *olp1*Δ mutants (TBL337), *olp1*Δ::*OLP1* complementation strains (TBL349), and *OLP1* overexpression strain *OLP1*^OE^ (TBL353). Both the *olp1*Δ mutant and *OLP1*^OE^ strain are less virulent than the wild-type H99 strain. ns, not significant; ****, *p* ≤ 0.0001 (determined by log-rank [Mantel–Cox] test). (**B**) Organs from five mice infected with H99, the *olp1*Δ mutants, *olp1*Δ::*OLP1* complemented strains, and *OLP1*^OE^ strain were dissected for colony-forming unit (CFU) counting at the end time point of infection. The data shown are the mean ± SD for values from five animals. *, *p* ≤ 0.05; **, *p* ≤ 0.01; ****, *p* ≤ 0.0001 (determined by Mann–Whitney test). (**C**) Hematoxylin and eosin (H&E)-stained slides were prepared from cross-sections of infected organs at the endpoint of the experiment and visualized by light microscopy. The cryptococcal cells are indicated by arrows. Brain: Bar = 20 μm, Lung: Bar = 20 μm, Spleen: Bar = 20 μm.

**Figure 7 microorganisms-08-01730-f007:**
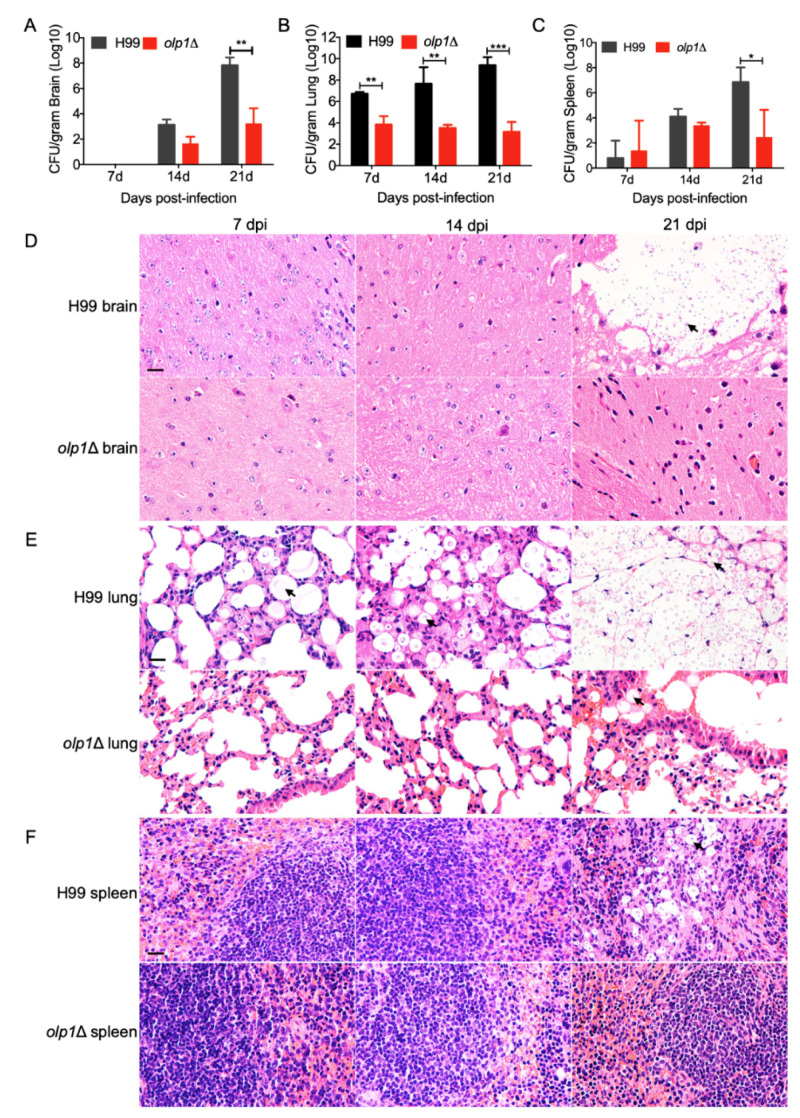
Progression of *olp1*Δ mutants infection in organs of infected animals. (**A**) Brains from five mice infected with H99 and the *olp1*Δ mutants were isolated at 7, 14, and 21 days postinfection (DPI), respectively. CFU per gram of fresh organ was measured in brain homogenates. (**B**) Lungs from five mice infected with H99 and the *olp1*Δ mutants were isolated at 7, 14, and 21 DPI, respectively. (**C**) Spleens from five mice infected with H99 and the *olp1*Δ mutants were isolated at 7, 14, and 21 DPI, respectively. Each data point and the error bar indicate the mean and standard error of the mean for values from five animals. *, *p* ≤ 0.05, **, *p* ≤ 0.01, ***, *p* ≤ 0.001 (determined by Mann–Whitney test). (**D**–**F**) H&E-stained slides were prepared from cross-sections of infected brains (**D**), lungs (**E**), and spleens (**F**) at 7, 14, and 21 DPI, respectively, and visualized by light microscopy. Bar, 20 μm. Arrows indicate the cryptococcal cells.

**Table 1 microorganisms-08-01730-t001:** Strains and plasmids used in this study.

Strains/Plasmids	Genotype/Properties	Source/Reference
*C. neoformans*		
H99	*MAT*α	[43]
KN99**a**	*MAT* **a**	[44]
TBL101	*MAT*α *P_ACTIN_-Nop1-mCherry*::*NAT*	[45]
TBL102	*MAT***a***P_ACTIN_-Nop1-mCherry*::*NAT*	[45]
TBL337	*MAT*α *olp1*::*NEO*	In this study
TBL348	*MAT***a***olp1*::*NEO*	In this study
TBL349	*MAT*α *olp1*::*NEO OLP1*::*NAT*	In this study
TBL352	*MAT**a** olp1*::*NEO OLP1*:: *NAT*	In this study
TBL353	*MAT*α *olp1*Δ::*NEO P_ACT1N_-OLP1-mCherry*::*NAT*	In this study
TBL354	*MAT***a***olp1*Δ::*NEO P_ACT1N_-OLP1-mCherry*::*NAT*	In this study
TBL371	*MAT*α *olp1*Δ::*NEO P_ACTIN_-Nop1-mCherry*::*NAT*	In this study
TBL372	*MAT***a***olp1*Δ::*NEO P_ACTIN_-Nop1-mCherry*::*NAT*	In this study
TBL373	*MAT*α *olp1*Δ::*NEO P_OLP1_-OLP1-mCherry*::*NAT*	In this study
TBL374	*MAT***a***olp1*Δ::*NEO P_OLP1_-OLP1-mCherry*::*NAT*	In this study
Plasmids		
pJAF1	Amp^r^ Plasmid harboring *NEO* marker	[46]
pTBL1	Amp^r^ Plasmid harboring *NAT* marke	[45]
pTBL3	Amp^r^ Plasmid harboring *mCherry-GPD1* terminator and *NAT* marker	[45]
pTBL5	Amp^r^ Plasmid harboring *mCherry-GPD1* terminator and *NAT* marker under *ACTIN* promoter	[45]
pTBL68	Amp^r^ Vector for *P_ACTIN_-NOP1-mCherry-NAT* for nuclear positioning	[45]
pTBL85	Amp^r^ Vector for *P_ACTIN_-mCherry-OLP1* for Olp1 localization	In this study
pTBL118	Amp^r^ Vector for *P_OLP1_-OLP1-mCherry* for Olp1 localization	In this study
pTBL205	Amp^r^ Vector for *P_OLP1_-OLP1-NAT* for complementation	In this study

Note: **a**, **a** mating type.

**Table 2 microorganisms-08-01730-t002:** Primers used in this study.

Primers	Targeted Genes	Sequence (5′-3′)
TL17	M13F	GTAAAACGACGGCCAG
TL18	M13R	CAGGAAACAGCTATGAC
TL19	*NEO* split F	GGGCGCCCGGTTCTTTTTGTCA
TL20	*NEO* split R	TTGGTGGTCGAATGGGCAGGTAGC
TL59	*NEO* R4	TGTGGATGCTGGCGGAGGATA
TL67	*STE20A* α F	CCAAAAGCTGATGCTGTGGA
TL68	*STE20A* α R	AGGACATCTATAGCAGAT
TL69	*STE20A***a** F	TCCACTGGCAACCCTGCGAG
TL70	*STE20A***a** R	ATCAGAGACAGAGGAGCAAGAC
TL217	*GAPDH* qRT-PCR F	TGAGAAGGACCCTGCCAACA
TL218	*GAPDH* qRT-PCR R	ACTCCGGCTTGTAGGCATCAA
TL404	*OLP1* KO F1	CTCCCCAGACAAGCACATTCC
TL405	*OLP1* KO R1	CTGGCCGTCGTTTTACGACGCGTCTACACCACTCAGCAA
TL1253	*OLP1* KO F2	GTCATAGCTGTTTCCTGCAAGATTCTGTGCGTATGGTGTGC
TL1254	*OLP1* KO R2	GTTTGTTCTTTTGGCGGGTTTGAG
TL1255	*OLP1* KO F3	ATATGAATTGCTGCGTGTGACC
TL1256	*OLP1* KO R3	GCTTATGCTCCTTCTTCCAGTATT
TL410	*OLP1* KO F4	TCCAAAGAAGAAGACAGCAACC
TL515	*OLP1* mCherry F	TTAGTAAACTCGCCCAACATGTCTGGATCCATGCCTATTCACACTCTTGCTTC
TL516	*OLP1* mCherry R	CTTGCTCACCATTCTAGAACTAGTGGATCCTTTGCCCTCTGGCTTGGTTCTG
TL539	*OLP1* QPCR F1	TACGAGCCTCTCGACGATAC
TL540	*OLP1* QPCR R1	TCTGGCTTGGTTCTGTCTTTAC
TL556	*OLP1* Comp F	GATATCGAATTCCTGCAGCCCGGGGGATCCTCCAAAGAAGAAGACAGCAACCTA
TL1257	*OLP1* Comp R	CGGTGGCGGCCGCTCTAGAACTAGTGGATCTTTGCCTACAGGATTTTGGTCACT
TL727	*OLP1 Pro-OLP1* F1	TCGACGGTATCGATAAGCTTCGCCGACGACCAAGATACAG
TL729	*OLP1 Pro-OLP1* R1	ATTCTAGAACTAGTGGATCCTTTGCCCTCTGGCTTGGTTCTGTC

Note: **a**, **a** mating type.

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
