# Peer review of "The Role of Oxidoreductase-Like Protein Olp1 in Sexual Reproduction and Virulence of Cryptococcus neoformans"

_microorganisms, 2020, doi:10.3390/microorganisms8111730_

Round 1
Reviewer 1 Report
This is a resubmission and the authors have largely answered my concerns by altering language in the original draft. However, I do have one remaining issue and a few minor comments that I noticed this time.
As I (and the other reviewer) said previously, I think the qPCR data is being misinterpreted. As they respond, their phenotypic data suggests that there may be a sporulation issue (likely a failure to undergo meiosis in the basidium) but their qPCR data definitely does not point in that direction. Thus when they say in the revised manuscript "The results of qRT-PCR showed that, compared to the 0 h time point, the transcription level of OLP1 was down-regulated during mating, especially after 48 h, indicating Olp1 may play a role in later stages of mating after cell fusion (Fig. 2A)" it is confusing to the reader and presents an illogical interpretation of the data, especially without a time course of mating appearing at these time points as well. As before, I think this language should be changed.
Minor comments:
Table 1B is poorly aligned from column to column and is a bit hard to read.
Line 293-299: Sorry I missed this the first time, but it is generally convention when describing a fluorescently tagged allele to place the name of the tag on the side of the allele name where it is fused in the real protein. Ie., an N-terminal tag becomes mCherry-Nop1 and a c-terminal becomes Nop1-mCherry. This section conflicts between those allele names and the description of the Nop1 as N-terminal tagged.
Line 414: Also sorry I missed this the first time, but OLP1 was not essential for growth on lithium supplemented plates, those mutants were just more sensitive to lithium chloride.
Line 429-439: Nomenclature in this new paragraph is not consistent. Gene and species names should be in italics, deletions should be lower case, etc.
Reviewer 2 Report
My congratulations to the authors for a very interesting work. The finding of a new oxidoreductase-like protein that demonstrates such an important role in the life cycle of pathogenic Cryptococcus yeasts is of remarkable interest. Therefore, I consider that the manuscript has to be published. Nevertheless, it needs some review in aspects related to the explanation of methods and presentation of results, as well as other minor details that are commented below.
In general, please try to avoid the use of Saxon genitive.
The main problems are in the Material and Methods section.
- The detailed explanations about how the experimental procedures were performed, have to be given in the material and methods section instead of in the results section. Now many of them appear in the text (paragraph 3.5 page 11) and in the figure legends. In fact, all figure legends except Fig 1 have an extensive explanation about procedures that have to be moved to materials and methods.
- Please provide the meaning of the abbreviations the first time they are used (YDP page 3, line 99; DME page 5, line 155; IP pull down; LC/MS page 6 lines 176-177; etc)
- Also add a comment or simple explanation when some specific products, genes or strains are used like mCherry gene in the subsection 2.2, in the text or the head of table 1 “fluorescent gene” has to be added. In page 5, line 127, please add “antibiotic” to the product G418. These terms could not be so familiar to other readers and having the keyword (fluorescent, antibiotic) makes the text more understandable.
- Fungal load on infected organs (page 6 line 173). The incubation period for Cryptococcus is 48 hours, and this is considered in most of the work, except in the experiment in which the fungal load of the infected organs of the mice is considered. Is there a reason for this change in procedure?
Results:
Apart from what was mentioned in the above paragraph (point 1), Figure 3 has many details that are not commented in the figure legend. All the details included in the figure have to explained in the legend. Please modify the figure (simplify) or appropriately complete the legend. The same for the word “DIC” in figures 2 and 5.
Discussion
The high fungal burden of the olp1∆ strain of Cryptococcus in the brain of infected mice at the endpoint, compared to the results obtained with the same strain 21 days after infection, deserves to be more commented. It seems that this strain needs more time than the wild type to reach a similar fungal burden at the end of the infection, in contrast to what appeared in lungs and spleen. Do you have any suggestion to explain the different behavior of the yeast in CSN? Minor problems:
Page 6, line 185. Something is missing after (94%). Maybe the word with or within?
Figure 3 legend: Line 242: Probably you mean 3B instead of 4B (the same in line 247). In Line 245, you probably mean olp1∆ instead of cop1∆ (the same mistake in Figure 5, line 284).
Figure 4 legend. The numbering of the comments is incorrect. Comments for 4B are lacking and the others are mixed up (4B corresponds to 4C and this one to 4D).
Page 13 line 352, it is probably 6C instead of 7C
In reference number 7, the name of the journal is mSphere instead of Msphere.
Author Response
Please see the attachment.

This manuscript is a resubmission of an earlier submission. The following is a list of the peer review reports and author responses from that submission.
Round 1
Reviewer 1 Report
In the manuscript entitled “The role of oxidoreductase-like protein Cop1 in sexual reproduction and virulence of Cryptococcus neoformans” by Yu et al., the authors identified an oxidoreductase-like protein, Cop1, in the human fungal pathogen Cryptococcus neoformans. By analyzing the deletion strains, complementation strains, as well as the overexpression strains the authors showed that the COP1 gene is required for 1) vegetative fitness in the presence of stresses such as LiCl and 5-FC; 2) robust hyphal growth, basidia maturation, and meiosis and sporulation during sexual reproduction; and 3) virulence in animal hosts. Overall the manuscript is clearly written and the data are well presented.
General/major comments
- The manuscript could benefit from more discussion to provide better interpretation of their data. For example the deletion strain of the oxidoreductas-like protein is inviable in S. cerevisiae, but it doesn’t appear to be essential in Cryptococcus. Are there any other putative oxidoreductase-like proteins in the Cryptococcus genome? If not, how do the cells survive without it and appear to be healthy under most of the conditions tested? Also, why would an oxidoreductase-like protein be required for sporulation? Do the authors have any hypotheses? It would be nice if the authors can add some discussion on these topics.
- In the C. neoformans genome, the COP1 gene (CNAG_00472) overlaps with the 3’UTR of the neighboring gene CNAG_00473 that encodes “oligosaccharyltransferase complex subunit epsilon”, which appears to be the homolog of the OST2 gene in yeast and whose deletion in yeast shows similar phenotypes (e.g. growth defect and increased sensitivity to chemical and oxidative stresses) as those described in this paper. Based on the primer information provided in the paper, it seems that the deletion of the COP1 gene also removed the overlapping 3’UTR region of the CNAG_00473 gene. Have the authors checked the function of the CNAG_00473 gene in their deletion strain to see if it is also affected? It would also be helpful to provide more information on the 3.9 kb region that was used to construct the complementation strain and see if it contains the CNAG_00473 ORF.
Specific/minor comments
- It would be nice to also show that reciprocal blast using yeast oxidoreductase also has the gene CNAG_00472 as the top and only hit in the Cryptococcus genome (Figure 1B).
- Lines 185-186. I think suggesting the Cop1 protein is C. neoformans-specific is a bit too strong, as that implies the gene does not have orthologs in other species, which might not be true.
- Lines 200-202. Figure 2A shows that the expression of the COP1 gene is highly reduced after 48 hours of mating. How does this indicate the gene plays a role in later stages of mating? If anything, shouldn’t this suggest the opposite?
- Figure 4 and Lines 264-265. It appears that the cop1 deletion strain is also more sensitive to SDS and Congo Red. Also, the callout in this paragraph should be “Fig. 4C” instead of “Fig. 5C” and “Fig. 5D”.
- The scale bars in Figure 5 appear to be off. For example, the three images of the basidia in Fig. 5B appear to be in same scale but the scale bar in the first image is shorter than the other two. Also, between Fig. 5A and 5B, if the scale bars are correct, then the basidia are >10 um in size in 5A but <10 um in 5B. I would recommend the authors to double check this to make sure.
- In Figure 6C, the label for the complementation strain has been cut off.
- Lines 397-398. I think the suggestion that the COP1 gene is involved in a novel virulence control mechanism is a bit premature. Could the observed attenuated virulence be due to the fact that the deletion strains are less fit compared to the WT parental strains under stress condition?
- Regarding Figure 6B, it is interesting that the deletion strain has more CFUs in the brain and less CFUs in the lung and spleen when compared to the WT and complementation strains. Does this imply the mice died from different causes (e.g. cryptococcal meningitis vs. pulmonary infection) when inoculated with the deletion strain? What if instead of the end point of infection the authors check the CFU at a certain time point, for example 2 weeks post infection? Also, do CFUs recovered from mice still show the same fitness defects as the initial deletion strains used for inoculation under stress conditions?
- Supplemental Figure 1B, the label “cor1” should be “cop1”.
- There are several places in the text with extra/missing spaces, such as in lines 228, 234, 271, and 337.
Reviewer 2 Report
This paper describes the characterization of an oxidoreductase gene in Cryptococcus neoformans that the authors have denoted COP1. They make a deletion, complemented, and overexpression strain of COP1 and demonstrate that COP1 plays a role in the ability to undergo meiosis during sexual reproduction, cause disease in a murine model of infection and resistance to the antifungal 5FC and to Lithium Chloride. This is a solid characterization of the function of Cop1. However, while the data is largely solid, there are numerous issues with interpretation of that data.
Concerns:
I think COP1 has the potential to be a confusing name- it immediately brings to mind the COP1 of cerevisiae, a relatively well known component of vesicle transport and not the ortholog of the COP1 named here.
Line 185: The authors suggest that COP1 may be a C. neoformans specific protein. This is definitely not true. At the very least, it is present in C. deuterogatti as well (in the table in Figure 1B with an e-value approaching 0). But in addition the similarity to S. cerevisiae PBR1 (which is also identified as the ortholog by FungiDB) suggests this is found much more broadly.
The neighbor joining phylogeny in Figure 1 is not very meaningful and lacks any kind of bootstrap support. I don't think this phylogeny is very important to the paper and could just be removed.
Line 199: The authors say that the reduction of expression of COP1 as time progresses post mating indicates that Cop1 may play a role in later stages of mating after cell fusion. This is the exact opposite of how you might interpret this qpcr data. If anything, a gene decreasing in expression over time would suggest less of a role over time.
Lines 222-225: Expression of COP1 throughout sexual reproduction (especially at similar or lower levels than expression in typical mitotic cells) does not mean or even suggest that it plays an essential role in sexual reproduction. Later experiments do demonstrate this, but this experiment does not.
Line 351: Again, Cop1 is not essential for fungal virulence. If it was, no mice would have died in this experiment. It is important for it, but not essential.
Line 392: Essential for sexual reproduction yes, but not for virulence.
Line 398: More likely than a novel virulence control system is that the Cryptococcus strains are just sick/stressed lacking COP1- they do have a growth defect even on rich media.
Minor comments:
Some tables and figures are cutoff throughout the manuscript.
In general the language is good but could use another pass to clean up some remaining grammar issues.